# The Secretome of Mesenchymal Stromal Cells in Treating Intracerebral Hemorrhage: The First Step to Bedside

**DOI:** 10.3390/pharmaceutics15061608

**Published:** 2023-05-29

**Authors:** Stalik Dzhauari, Nataliya Basalova, Alexandra Primak, Vadim Balabanyan, Anastasia Efimenko, Mariya Skryabina, Vladimir Popov, Arkadiy Velichko, Kirill Bozov, Zhanna Akopyan, Pavel Malkov, Dmitry Stambolsky, Vsevolod Tkachuk, Maxim Karagyaur

**Affiliations:** 1Faculty of Medicine, Lomonosov Moscow State University, 27/1 Lomonosovsky Ave., 119192 Moscow, Russia; stalik.djauari@yandex.ru (S.D.); primak.msu@mail.ru (A.P.); bal.pharm@mail.ru (V.B.); aefimenko@mc.msu.ru (A.E.); skrebbka@gmail.com (M.S.); galiantus@gmail.com (V.P.); velichko.arkady@gmail.com (A.V.); kir-bozov@yandex.ru (K.B.); zhanna.fbm@gmail.com (Z.A.); pmalkov@mc.msu.ru (P.M.); tkachuk@fbm.msu.ru (V.T.); 2Institute for Regenerative Medicine, Medical Research and Education Center, Lomonosov Moscow State University, 27/10 Lomonosovsky Ave., 119192 Moscow, Russia; natalia_ba@mail.ru; 3Medical Research and Education Center, Lomonosov Moscow State University, 27/10, Lomonosovsky Ave., 119192 Moscow, Russia; stambolsky@mail.ru

**Keywords:** multipotent mesenchymal stromal cells (MSC), secretome, intracerebral hemorrhage, stroke, route of administration, dose–response relationship, door-to-treatment time, course treatment, aged animals

## Abstract

Intracerebral hemorrhage is an unmet medical need that often leads to the disability and death of a patient. The lack of effective treatments for intracerebral hemorrhage makes it necessary to look for them. Previously, in our proof-of-concept study (Karagyaur M et al. Pharmaceutics, 2021), we have shown that the secretome of multipotent mesenchymal stromal cells (MSC) provides neuroprotection of the brain in a model of intracerebral hemorrhage in rats. Here, we have conducted a systematic study of the therapeutic potential of the MSC secretome in the model of hemorrhagic stroke and provided answers to the questions that need to be addressed in order to translate the secretome-based drug into clinical practice: routes and multiplicity of administration, optimal dose and door-to-treatment time. We have found that MSC secretome reveals prominent neuroprotective activity when administered intranasally or intravenously within 1–3 h after hemorrhage modeling, even in aged rats, and its multiple injections (even within 48 h) are able to reduce the delayed negative effects of hemorrhagic stroke. To our knowledge, this study provides the first systematic investigation of the therapeutic activity of a biomedical MSC-based cell-free drug in intracerebral hemorrhage and is an integral part of its preclinical studies.

## 1. Introduction

Stroke is a severe cerebrovascular disease accompanied by the death of part of the brain, which may lead to the disability or death of the patient. Annually, 12.2 million new strokes are registered globally [1], and intracerebral hemorrhages (hemorrhagic stroke) account for about 28% of the total number of strokes. Despite the lower prevalence compared to ischemic stroke, intracerebral hemorrhage is characterized by high mortality (up to 50%) and disability among survivors (up to 85%) [1,2].

Most pathogenetic mechanisms of intracerebral hemorrhage are known; moreover, they take part in other types of acute damage to the brain tissue [3,4]. Understanding these mechanisms makes it possible to correct them and reduce the severity of brain damage. In order to provide a pronounced therapeutic effect, the treatment should be targeted at several main pathogenetic mechanisms of intracerebral hemorrhage, and the medicine used should contain several functionally-complement active substances, or its active substance should have pleiotropic effects. Previously, we have shown that the secretome of mesenchymal stromal cells (MSCs) meets these requirements and provides a prominent neurotrophic effect in the model of intracerebral hemorrhage [5]. To date, MSCs have been found in almost all organs and tissues [6,7]. It is believed that MSCs are the key endogenous coordinators of tissue renewal and regeneration; herewith, they realize most of their protective and pro-regenerative effects via the production of angiogenic and neurotrophic factors, matrix proteins, and anti-inflammatory molecules [8,9]. The entire complex of protein and non-protein molecules (including extracellular vesicles) produced by MSCs comprises MSC secretome, which, according to the results of many studies, stimulates regeneration and exhibits pronounced therapeutic activity in a number of diseases [10,11].

In our previously published proof-of-concept study, we demonstrated that MSC secretome, when injected into the hemorrhage focus, promotes the survival of experimental animals and reduces the severity of neurological deficits [5]. However, the translation of the applied approach to the clinic is difficult for a number of reasons, despite the shown effectiveness. Above all, secretome administration directly into the area of damage is practically unrealizable in the clinic, whereas it is not obvious if MSC secretome retains its neuroprotective activity when administered using an alternative, clinically relevant route of administration. Moreover, we have not studied the optimal door-to-treatment time previously, and the time after a stroke that the secretome retains its neuroprotective effect remains unknown. In this study, we tried to answer these and other questions that inevitably arise during the preclinical study of a promising drug and its translation into the clinic. To our knowledge, the results obtained provide the first systematic study of the therapeutic activity of a biomedical cell-free drug in intracerebral hemorrhage.

## 2. Materials and Methods

### 2.1. Animals

Rats of two ages were used: 3.0–3.5 month-old rats (weighing 300–350 g) for most experiments, and 10–11 month-old rats (weighing 450–560 g; BMI 0.69–0.86) for assessing the neuroprotective activity of MSC secretome in aged animals [12,13]. Animals were housed and used for experimental procedures in full compliance with Directive 2010/63/EU and the recommendations of the Bioethics Committee of Lomonosov MSU (protocols #3.4 (21 March 2021) and #3.5 (17 March 2022)).

### 2.2. Cell Culture

Human MSCs derived from adipose tissue of healthy donors (*n* = 3) were obtained from the biobank of the Institute for Regenerative Medicine, Lomonosov MSU, collection ID: MSU_MSC_AD (https://human.depo.msu.ru (accessed on 23 May 2023)) and cultured in the medium supporting the growth of undifferentiated mesenchymal progenitor cells (Advance Stem Cell Basal Medium, HyClone, South Logan, UT, USA) containing 10% of a growth factor supplement (Advance Stem Cell Growth Supplement, HyClone, South Logan, UT, USA) and 100 U/mL of penicillin/streptomycin (Gibco, Grand Island, NY, USA). All procedures performed with tissue samples from patients were in accordance with the Declaration of Helsinki and approved by the Ethics Committee of Lomonosov Moscow State University (IRB00010587), protocol #4 (2018).

Obtained MSCs were characterized as being plastic adherent, expressing CD73, CD90, and CD105, lacking the expression of hematopoietic and endothelial markers CD14, CD19, CD34, CD45, and HLA-DR, and capable of in vitro differentiation into adipocyte, chondrocyte, and osteoblast lineages which met the criteria set by the International Society for Cellular Therapy (ISCT) [14,15]. The medium was changed every 3–4 days. All experiments were performed with cells within 5 passages.

### 2.3. Manufacturing of Cell Secretomes

MSC-conditioned medium containing components of human MSC secretome was obtained according to the previously published protocol [16]. Briefly, subconfluent pooled human MSC cultures at passages 4–5 were thoroughly washed with Hanks solution (PanEko, Moscow, Russia) and were then cultured for 7 days in DMEM containing low glucose (DMEM-LG), GlutaMAX™ Supplement, pyruvate (DMEM-LG; Gibco, Grand Island, NY, USA), and 100 U/mL penicillin/streptomycin. Then, the medium of human MSC was removed and centrifuged for 10 min at 300× *g* to remove cell debris, and then it was concentrated 5, 10, 25, or 50 fold (MSC5x, MSC10x, MSC25x, and MSC50x groups) using a centrifugal ultra-filter with 10-kDa molecular weight cut-off (MWCO; Merck, Darmstadt, Germany). Normalization of the secretome composition was carried out according to BDNF (brain-derived neurotrophic factor) as one of the main neuroprotective components of the MSC secretome [17,18,19]. Non-concentrated MSC secretome (MSC1x) contained 3 ± 1.2 ng/mL of BDNF.

### 2.4. Intracerebral Hemorrhage Modeling and Secretome Administration

The study was carried out on a model of intracerebral hemorrhage into the inner capsule of the right hemisphere, according to Makarenko et al. [20]. The inner capsule contains many neural tracts that connect the cortex to the basal ganglia, nuclei of the spinal cord, and cranial nerves. Therefore, injury to the inner capsule causes various neurological deficits, and their severity correlates with brain injury severity.

The study included 152 animals: 30 rats for the root of administration study, 35 rats for the door-to-treatment time study, 30 rats for the dose-dependence study, 27 rats for the course-treating study, and 20 rats for the aged-rats treating study. To model the hemorrhage, rats were anesthetized with a solution containing 2% Zoletil^®^ 50 (VirBac, Hamilton, New Zealand) and 1.5% xylazine (InterChemie, Venray, The Netherlands) in saline at a dose of 1 mL/kg body weight and positioned in a stereotaxic frame. The scalp and skull aponeurosis were dissected, and the cranium was perforated (bregma −2.0 mm, lateral 3.5 mm) [21]. Local brain tissue was destroyed, and the superior cerebral veins on the corresponding side were injured to model a bleed. To standardize the volume of spilled blood, 20 µL of autologous blood obtained from the hypoglossal vein was slowly injected into the injury focus, and the wound was sutured. For choosing the optimal administration route of MSC secretome for brain neuroprotection, the effectiveness of intracerebral, intravenous, intrathecal, and intranasal administration routes was compared. For this, 10× MSC secretome or DMEM-LG medium (Control group) was administered to anesthetized animals 5–10 min after modeling the intracerebral hemorrhage. Intracerebral injections of 40 µL of 10× MSC secretome or DMEM-LG medium were performed as described earlier [5] right into the hemorrhage site, 5 min after blood administration. Intravenous injections of 100 µL of 10× MSC secretome were performed into the tail vein. Intrathecal injections of 100 µL of 10× MSC secretome were performed between L5 and S1. For intranasal administration, 40 µL of 10× MSC secretome was injected into each nasal passage (80 µL in total). For intranasal secretome administration, rats with the hemorrhage were placed onto their backs and stayed in such position for 25–30 min to let the secretome absorb. The volume of 10× MSC secretome administered was dictated by the maximum capacity of safe and effective injection volume in each route.

During the determination of the door-to-treatment time, the blinded investigator injected 10× MSC secretome or DMEM-LG medium (100 µL each) into the tail vein or into nasal passages (40 µL/nasal passage) of anesthetized animals in 1 h, 3 h, or 6 h after modeling the intracerebral hemorrhage. During the evaluation of course-administration MSC secretome effectiveness, 100 µL of 10× MSC secretome or DMEM-LG medium (Control group) was injected into the tail vein of anesthetized animals 1, 48, 96, and 144 h after modeling the intracerebral hemorrhage. In other studies, 100 µL of MSC secretome or DMEM-LG medium was injected into the tail vein of anesthetized animals one hour after modeling the intracerebral hemorrhage. Three rats who died within 24 h after intracerebral hemorrhage modeling (course-treating and aged-rats treating studies) were excluded from the study.

### 2.5. Neurological Status Assessment

Most animals were observed for 14 days following the intracerebral hemorrhage, with daily recording of rat deaths and neurological status assessment at 3 and 10 days following the hemorrhage. A total of 10 animals from the course-treating study (5 each from the DMEM-LG and MSC secretome groups) were followed up to 28 days after modeling the intracerebral hemorrhage to elucidate possible delayed effects of the use of MSC secretome and their possible reversibility. For neurological status assessment, we used the stroke index McGraw scale, modified for rodents by I.V. Ganushkina [22,23]. Briefly, “visually healthy animals” had no signs of neurological deficit. The animals with signs of lethargy, limb weakness, tremor, ptosis, and/or semi-ptosis were considered “slightly affected”. The animals demonstrating signs of paresis and/or paralysis of the limbs, impaired coordination, or in a coma were considered “severely affected”. The researchers conducting the neurological testing were blinded to each rat’s treatment. The brains of animals that died 24 h or later after modeling the intracerebral hemorrhage were removed, fixed in 4% formaldehyde solution, and included in the MRI and histological examination.

### 2.6. MRI

MR images were obtained 14 days (or 28 days for ten animals from the course-treating study) following the intracerebral hemorrhage using the Clinscan 7T system (Bruker Biospin, Billerica, MA, USA) equipped with the rat brain surface coil with TurboS Spin Echo sequence and fat suppression. The coronary projections were acquired using the following parameters: TR (Repetition Time) = 5220 ms; TE (Time to Echo) = 53 ms; echo train length = 9; base resolution 230 × 320; FoV (Field-of-View) = 32 × 40 mm; slice thickness = 0.5 mm; spacing between the slices = 0.75 mm. The trans-versal projections were acquired using the following parameters: TR = 4000 ms; TE = 40 ms; echo train length = 9; base resolution 288 × 320; FoV = 40 × 40 mm; slice thickness = 0.5 mm; spacing between the slices = 0.6 mm.

The volume of brain lesions, according to MRI data, was estimated as the product of the sum of the areas of damaged brain tissue on individual MRI slices and the sum of the slice thickness (0.5 mm) and the distance between slices (0.75 mm). The area of the damaged brain tissue on individual MRI slices was determined using the Radiant DICOM Viewer 2020.1.1 program (Medixant, Poznan, Poland).

### 2.7. Histochemistry

For histological studies, the rats were euthanized on day 14 (or day 28 for ten animals from the course-treating study) following the intracerebral hemorrhage by exposure to gradually increasing concentrations of CO_2_. The brain was removed, fixed in a 4% formaldehyde solution, and embedded in paraffin. The brain slices containing the injury focus were dewaxed, and the slices were stained with hematoxylin-eosin. The specimens were examined with a Leica DM600 microscope equipped with a DFC420C camera (Leica Microsystems GmbH, Wetzlar, Germany), using representative fields of view to obtain photographs. Image processing and analysis were performed using LasX software (Leica Microsystems GmbH, Wetzlar, Germany) and Fiji.

### 2.8. Statistical Analysis

Statistical analysis was performed using SigmaPlot11.0 software (Systat Software, Inc.; Erkrath, Germany). Numerical data were assessed for normality of distribution using the Kolmogorov–Smirnov test. Differences between the treatment and Control groups were analyzed using Student’s *t*-test or analysis of variance (ANOVA) on ranks (Dunn’s test), depending on whether the data were normally distributed. Data are expressed as the median (25%; 75%). We considered differences to be significant when *p* < 0.05.

To analyze the categorical data (neurological outcome after stroke), Fisher’s exact test was used: the treatment groups were compared to the Control group pairwise. Since Fisher’s exact test only supports 2 × 2 massives (χ^2^ is not applicable for small groups), the proportions of animals were combined into two groups: “visually healthy animals” + “slightly affected”, “severely affected” + “dead”. Different points in time were compared separately.

## 3. Results

### 3.1. Intravenous and Intranasal Administration of MSC Secretome Provides Neuroprotection in the Model of Intracerebral Hemorrhage

We propose the application of the found neuroprotective effect of MSC secretome in clinical practice. For this goal, the first task is to find an accessible, clinically relevant route of secretome administration to treat intracerebral hematoma. To do this, we studied the effectiveness of MSC secretome being administered intravenously, intranasally, or intrathecally, in comparison with the previously described group of MSC secretome administered right into the focus of hemorrhage.

The results of the study reveal that MSC secretome retains its neuroprotective effect in the model of intracerebral hemorrhage, even when administered intravenously or intranasally. Thus, in these experimental animal groups, we observed a lower severity of hematoma-induced neurological disorders than in the Control group, where DMEM-LG medium was administered intravenously (Figure 1A), although no statistically significant difference was observed due to the small sample size. Intravenous and intranasal administration of 10× MSC secretome decreased the lesion volume by 2 and 1.3 times—down to 108 (75; 172) and 173 (147; 175) mm^3^, respectively, compared to the Control group (intravenous administration of DMEM-LG) with an average lesion volume of 218 (201; 233) mm^3^ (Figure 1B,C). Moreover, according to our MRI and histological data, intravenous administration of MSC secretome reveals a more prominent neuroprotective effect than the secretome injected directly into the hemorrhage focus. Apparently, this is due to the gradual distribution of secretome components from the bloodstream into the injury focus and, as a result, less microglia cell activation within the damage area (or probably due to another, yet unidentified, mechanism). Intrathecal administration of MSC secretome did not cause any observable improvement in the lesion size or severity of neurological disorders. The histological data obtained are consistent with the MRI results (Figure 1B).

### 3.2. Door-to-Treatment Time in the Model of Intracerebral Hemorrhage Ranges from 1 to 3 h for Intravenous and Intranasal Administration of MSC Secretome

Since it is known that the effectiveness of therapy for acute brain injuries largely depends on the time of treatment initiation after injury, it was necessary to establish the effective door-to-treatment time for MSC secretome therapy in the intracerebral hemorrhage model. We studied the door-to-treatment time (1, 3, and 6 h after the intracerebral hemorrhage modeling) for the routes of MSC secretome administration that turned out to be the most neuroprotective, according to the previous study stage—intravenous and intranasal. The greatest neuroprotective effect was observed for intravenous administration of 10× MSC secretome 1 h after the modeling of intracerebral hemorrhage: all five experimental animals in this group had no signs of neurological disorders during the entire observation period (14 days) (Figure 2A) with an average brain lesion volume of 86 (62; 109) mm^3^ (Figure 2B,C). In the Control group (intravenous administration of DMEM-LG), at least 2 animals had neurological disorders during the observation period with an average brain lesion volume of 205 (189; 232) mm^3^ (*p* < 0.05; *n* = 5). Increasing the time interval of drug administration after modeling the hemorrhage decreased the neuroprotective activity of the MSC secretome. Thus, the brain lesion volume in groups with intravenous administration of 10× MSC secretome 3 and 6 h after modeling the hemorrhage reached 168 (162; 203) and 216 (203; 224) mm^3^, respectively. The neuroprotective activity of MSC secretome in rats with intranasal administration decreased with an increase in the door-to-treatment time to 1–6 h and did not differ significantly from that in the Control group. The results of the histological study coincided with the data from the MRI examination (Figure 2B).

Since the highest neuroprotective activity of the MSC secretome was observed with its intravenous administration 1 h after modeling the intracerebral hemorrhage, this combination of the route of administration and door-to-treatment time was used when titrating the dose, studying the effectiveness of course treatment and the neuroprotective activity of the MSC secretome in aged rats, etc.

### 3.3. MSC Secretome, 5- and 10-Fold Concentrated, Reveals the Optimal Neuroprotective Activity

The next step in studying the neuroprotective properties of the MSC secretome was to establish its optimal “dosage”, i.e., the concentration of MSC secretome that reveals the highest neuroprotective activity when administered at a constant volume (intravenous administration, 100 µL). Normalization of the MSC secretome composition was carried out according to one of the main neuroprotective components of the secretome—BDNF (brain-derived neurotrophic factor). For non-concentrated MSC secretome (1×), BDNF concentrations range within 3 ± 1.2 ng/mL. For MSC secretome samples concentrated 5-, 10-, 25-, and 50-fold (5×, 10×, 25×, 50×, respectively), BDNF concentration reached 15 ng/mL, 30 ng/mL, 75 ng/mL, and 150 ng/mL, respectively. We used DMEM-LG medium concentrated 50-fold as a Control group in this study.

The study demonstrated that MSC secretome, concentrated 5-, 10-, 25-, and 50-fold, revealed the neuroprotective effect and reduced the severity of neurological disorders in operated rats, compared to the Control group, although the differences observed are not statistically significant (due to the small sample). According to the MRI study, the MSC secretome, concentrated 5- and 10-fold times, had the greatest neuroprotective effect. Thus, 5× or 10× MSC secretome being administered intravenously 1 h after the modeling of intracerebral hemorrhage reduced the volume of brain tissue lesion to 113 (93; 124) mm^3^ and 156 (104; 172) mm^3^, respectively, with an average lesion volume of 229 (199; 359) mm^3^ in the Control group. Non-concentrated 1× MSC secretome did not reveal any significant neuroprotective effect, which was consistent with the previously obtained data [5]. MSC secretome, concentrated 25- and 50-fold, on the contrary, increased the brain lesion volume to 299 (277; 330) mm^3^ and 347 (215; 474) mm^3^, respectively. The results of the histological study coincided with the data from the MRI examination (Figure 3B). For further studies, we used 10-fold concentrated MSC secretome (10×, with a BDNF content of 30 ng/mL) since this concentration revealed the most prominent neuroprotective activity for the brain tissue after acute injury.

### 3.4. Course Treatment with MSC Secretome Stalls the Pathogenesis of Intracerebral Hemorrhage and Provides a More Prominent Neuroprotective Effect than a Single Injection

According to the literature [3,4,24], pathogenesis of an acute brain lesion develops over time and results in secondary brain tissue damage due to the activation of microglia and the persistence of disturbing factors. Taking this into account, we assumed that repeated administrations of MSC secretome would mitigate the negative impact of damaging factors within the brain tissue and prevent its secondary damage. The results of the study confirmed this. Thus, course treatment with MSC secretome (intravenous administration 1, 48, 96, and 144 h after modeling the intracerebral hemorrhage) reduced the severity of neurological deficits compared to the Control rats receiving treatment with DMEM-LG (although no statistically significant difference was observed) (Figure 4A), and significantly reduced the brain lesion volume according to MRI. Rats that obtained a course treatment with MSC secretome (4 injections) had an average brain lesion volume of 36 (33; 77) mm^3^, while in the Control group, it was 198 (146; 207) mm^3^, and in rats after a single MSC secretome administration—122 (105; 177) mm^3^ (significant differences between each pair of groups, *p* < 0.05, *n* ≥ 5) (Figure 4B,C).

During this experiment, some animals were housed for 22 days after the last MSC secretome administration within the course treatment (for 28 days in total from the moment of modeling the intracerebral hemorrhage) to monitor possible delayed negative side effects of MSC secretome treatment. The results of this study revealed no negative long-term side effects of MSC secretome administration in experimental animals. At the same time, the neurological deficits and brain lesion volume (according to MRI data) in those rats did not differ significantly from that in rats course treated with MSC secretome and sacrificed 14 days after modeling the intracerebral hemorrhage and were significantly reduced compared to the corresponding parameters in rats from the respective Control group (Figure 4C). The results of the histological study coincided with the data from the MRI examination (Figure 4B).

### 3.5. MSC Secretome Provides Brain Neuroprotection in Aged Rats in the Model of Intracerebral Hemorrhage

Various forms of cerebrovascular diseases (including stroke) mostly occur in people in adulthood and are often associated with obesity, metabolic, or cardiovascular disorders. Since the effectiveness of neuroprotection and neuroregeneration processes may differ for a number of reasons in a young and mature organism, we decided to elucidate if MSC secretome would reveal its neuroprotective activity in the model of intracerebral hemorrhage in aged rats. The results of the study confirmed that a single intravenous administration of 100 µL of MSC secretome (10×) stimulated the survival and support of neurological functions of experimental animals and reduced the volume of the brain lesion in rats aged 10–11 months (Figure 4A,C). Thus, the brain lesion volume in old rats that received a single intravenous injection of MSC secretome (10×) was 156 (88; 196) mm^3^ vs. 219 (183; 334) mm^3^ in the Control group (*p* < 0.05; *n* ≥ 8) (Figure 4B,C). There were no significant differences in the neuroprotective efficiency of MSC secretome between young (3–3.5 months) and aged (10–11 months) rats. The results of the histological study coincided with the data from the MRI examination (Figure 4B).

## 4. Discussion

Intracerebral hemorrhage is a severe disease, and its symptoms are largely determined by the location and volume of the hemorrhage [2,25]. Hematomas of various sizes (both small and large) may lead to the most severe consequences, including the death of a person or animal. The volume of hemorrhage affects the course of pathological processes and determines the therapeutic tactics and the outcome after the stroke [25]. Thus, large hematomas are mainly lethal due to the displacement of the brain structures and compression of the brainstem. The therapeutic strategy of large hemorrhages must necessarily include surgical removal of the hematoma, and without it, therapy with a secretome (or another drug) is not effective and has little sense [25]. Smaller intracerebral hemorrhages also pose a danger. They may evoke damage to and death of neural cells [3,4] caused by ischemia, excitotoxicity, and toxic effects of blood (ferroptosis), as well as neuroinflammation, which leads to secondary injury to the brain tissue [26,27] and lesion expansion. If it affects the vital centers of the brain, it may also lead to the death of the affected person or animal.

Here, in this manuscript, we consider hematomas that do not displace the brain structures or compress the brainstem. If we draw a parallel between the model we use in rats with intracerebral hemorrhage in humans, then the administration of 20 μL of blood into the rat brain (average rat brain volume ~2.5 mL) corresponds to a hemorrhage in a person with a volume of up to 10 mL (average human brain volume ~1300 mL), which is quite a lot and may induce secondary brain damage without any significant mechanical stress to the brain structures.

Treating intracerebral hemorrhages requires the development of new therapeutic approaches [2,25] since existing drugs provide only symptomatic therapy and practically do not affect the main components of its pathogenesis, such as the death of neural cells (caused by ischemia, excitotoxicity, and ferroptosis), neuroinflammation, and secondary brain tissue injury [26,27]. It is obvious that almost no monocomponent drug is able to act on each of these pathogenesis mechanisms—this requires a complex effect of a combination of molecules with neuroprotective and anti-inflammatory activity. Moreover, it is not surprising that a combination of molecules within the secretome produced by mesenchymal stromal cells, the main regulators/coordinators of regeneration processes [8,9,10], has the necessary neuroprotective and anti-inflammatory activity.

Previously, we showed that MSC secretome administered into the lesion site of intracerebral hemorrhage promotes the survival of experimental animals and diminishes the severity of neurological disorders [5]; however, the proposed approach was practically not applicable for clinical use and did not answer a number of questions that need to be solved for translational research. In this study, we carried out a systematic analysis of the neurotrophic activity of MSC secretome at various routes, doses, and regimens of administration in the model of intracerebral hemorrhage in rats.

First of all, we have shown that the secretome of MSCs retains its neuroprotective activity using alternative clinically relevant routes of administration. Thus, intravenous and intranasal administration of human MSC secretome stimulates neuroprotection of the brain tissue in the model of intracerebral hemorrhage and does not increase the lesion focus volume compared to intracerebral administration [5]. Apparently, this is due to the fact that potentially immunogenic components of the human secretome being administered intravenously/intranasally exert not a simultaneous but a gradual and limited effect on immunocompetent cells of the brain (microglia). It cannot be ruled out that the potential immunogenicity of the secretome components during intravenous and intranasal administration is compensated by the presence of anti-inflammatory molecules (TGFb, indoleamine-2,3-dioxygenase, etc.) in MSC secretome [28,29].

The mechanism of the neuroprotective activity of MSC secretome administered intravenously was not established in this work. However, knowing the composition of the secretome, we assume that it may be due to a direct neuroprotective effect on neural cells or via the suppression of neuroinflammation, which is one of the main drivers of secondary brain injury after intracerebral hemorrhage.

A number of studies produced results in favor of the hypothesis of direct neuroprotection; they have shown that some proteins [30,31,32,33] and molecular complexes (microvesicles) [34,35] are able to penetrate the blood–brain barrier, providing neuroprotection and improving the outcome after stroke. Moreover, it has been demonstrated that the permeability of the blood–brain barrier can be increased under conditions of hemorrhage or local ischemia [36,37], which may lead to an increase in the supply of neuroprotective molecules to the injury site. A number of studies have established a correlation between increased concentrations of growth factors and neuroprotective miRNAs in the blood serum of patients with stroke and their better survival and neurological outcome [38,39,40,41,42,43]. Whether the high concentration of these molecules in blood plasma is a consequence of a brain injury or a constitutive feature (and an innate mechanism of endogenous neuroprotection) in these patients remains to be established.

It is known that MSC secretome is able to suppress the activation of T-lymphocyte and monocyte–macrophage cells [28,29,44]. It cannot be ruled out that the anti-inflammatory activity of the MSC secretome may mediate the indirect neuroprotection of the brain tissue. Presumably, MSC secretome, when administered intravenously, may increase the activation threshold of T cells and monocytes/macrophages that suppress their recruitment into the hemorrhage locus and prevents the progression of neuroinflammation and secondary damage. The mechanisms of neuroprotective activity of endovascular circulating growth factors in acute cerebrovascular accidents have yet to be established.

One of the widely used routes for drug administration in the pathologies of the nervous system is intrathecal administration [45,46]. However, in our study, intrathecal MSC secretome administration did not provide any neuroprotection to the brain tissue in the model of intracerebral hemorrhage, contrary to our expectations. Apparently, this is due to the rostral–caudal direction of cerebrospinal fluid flow [47] and the far distance between the injection site (L5–S1) and the hemorrhage locus. As a result, the secretome components did not reach the lesion site within the effective door-to-treatment time. With a high probability, this route of administration may be effectively used for the treatment of acute and chronic spinal cord injuries [45,46,48].

It is known that the effectiveness of therapy for acute brain injuries is largely determined by the promptness of neuroprotection [49,50]. In our study, we have shown that MSC secretome, when administered intravenously within 1 h after modeling the intracerebral hemorrhage, provides a better outcome in the experimental animals with the least severity of neurological disorders and the least brain lesion volume. This correlates with the literature data available, the so-called stroke golden hour—60 min after a stroke when the effectiveness of neuroprotection is maximum [51,52]. With an increase in the time interval between the modeling of intracerebral hemorrhage and intravenous administration of MSC secretome (from 1 to 6 h), the effectiveness of neuroprotection decreased, which correlated with literature data [49,50,51,52] and manifested in the worse neurological outcome and larger lesion volume in the brain tissue.

Spilled blood within an intracerebral hemorrhage accident triggers multiple pathogenetic mechanisms (ferroptosis, ischemia, neuroinflammation, etc.) that lead to secondary injury to the brain tissue [3,4,53], brain lesion expansion, and eventually may lead to the death of the patient. From this point of view, it is of particular interest to establish the possibility of MSC secretome to prevent the progression of secondary brain injury. We have found that repeated administrations of MSC secretome at 1, 48, 96, and 144 h after modeling the intracerebral hemorrhage led to a reduction in the brain lesion volume compared to the treatment outcome in the group with a single administration of MSC secretome 1 h after stroke modeling (according to MRI and histological examination). This confirms the fact that the pathogenesis of intracerebral hemorrhage and brain tissue injury are extended in time and demonstrates that MSC secretome is able to suppress the secondary brain injury in the pathogenesis of intracerebral hemorrhage, even beyond the so-called “stroke golden hour”. This largely extends the door-to-treatment time and potentially reveals new targets for pathogenetic therapy of intracerebral hemorrhages.

Determination of the optimal dose of a promising drug is one of the essential tasks of a preclinical study. We have shown that the optimal therapeutic effect is observed when using the MSC secretome 5- or 10-fold concentrated (with the BDNF content of 15 ng/mL and 30 ng/mL, respectively) being administered intravenously into the tail vein of rats one hour after the modeling of intracerebral hemorrhage. Non-concentrated MSC secretome (with the BDNF content of 3 ng/mL) did not reveal any significant therapeutic effect, and the administration of 100 μL of MSC secretome concentrated 25- or 50-fold (with the BDNF content of 75 ng/mL and 150 ng/mL, respectively), on the contrary, led to the enlargement of the brain tissue lesion according to MRI. Thus, in this study, we have established a range of neurotrophic activity of MSC secretome in treating intracerebral hemorrhages in rats: non-concentrated MSC secretome (1×) reveals no effect, while the 5- and 10-fold concentrated MSC secretome exerts the optimal neuroprotection, the use of 25-fold concentrated MSC secretome does not improve the outcome (reaches a “plateau”), and the 50-fold concentrated MSC secretome reveals toxic effects.

Presumably, this depends on the level of certain growth factors, neurotrophins or cytokines, which act in a rather narrow therapeutic range. Thus, it was previously shown that an excess of neurotrophic factors (or the predominance of their immature forms) leads to the signalization via the alternative p75 receptor and to the apoptosis of neural cells instead of their neuroprotection [54,55]. A similar dependence (increase in efficiency with dose, achievement of a “plateau”, and toxic effect) on the administration of different doses of MSCs or their secretome was repeatedly observed in other models and other routes of administration [56,57,58], which testifies in favor of universality of this phenomenon. The role of certain components of the secretome that determine the toxic effect in overdose, as well as the mechanisms of this phenomenon, have yet to be established; however, we do not exclude that it is based on an excess of growth (neurotrophic) factors or excessive stimulation of the immune system of the recipient. It is noteworthy that the DMEM-LG medium (Control group), 50-fold concentrated, did not exert any significant effect (protective or toxic) on the regeneration of the brain tissue in the model of an acute brain injury, even using a similar route and scheme of administration. This indicates that the observed neuroprotective and toxic effects are due to the specific action of certain components of the MSC secretome. It is obvious that before translating to the clinic, an additional determination of the effective, sufficient, and safe dose of the MSC secretome-based drug in humans will be required, and the data obtained in this study may serve as the basis for planning such a study.

Another important practical result of this study is that the MSC secretome provides neuroprotection in acute brain injury not only in young, healthy rats but also in overweight, mature rats (10–11 months old, weight up to 560 g). The metabolic status of aged rats was not assessed; however, according to a number of studies, an excess of the rat body mass index over 0.68 g/cm^2^ is associated with changes in the blood lipid profile and endocrine disorders [12,13]. The data obtained suggest that the use of MSC secretome may be a promising therapeutic approach for neuroprotection after acute brain injuries and cerebrovascular accidents, including intracerebral hemorrhage, in elderly patients with concomitant metabolic, endocrine, and possibly cardiovascular disorders.

## 5. Conclusions

In this study, we found that MSC secretome exerts a prominent neuroprotective effect in the model of intracerebral hemorrhage and retains its activity when administered using clinically relevant routes and modes; also, MSC secretome is able to stimulate neuroprotection in elderly animals. To our knowledge, this study is one of the first to systematically analyze the therapeutic activity of a biomedical cell-free promising drug (MSC secretome) in a model of intracerebral hemorrhage and is an integral part of its preclinical study.

The results obtained allow us to state that MSC secretome has a high neuroprotective activity, even under conditions close to clinical ones, and is a highly promising candidate for the development of a drug for the treatment of acute brain injuries, including intracerebral hemorrhage. A number of other aspects necessary for the translation of the MSC secretome-based drug into neurological clinical practice (such as the determination of the molecular mechanisms of neuroprotective activity, standardization of the secretome composition, safety, etc.) have yet to be investigated and developed. This will be performed in continuation of our study.

## Figures and Tables

**Figure 1 pharmaceutics-15-01608-f001:**
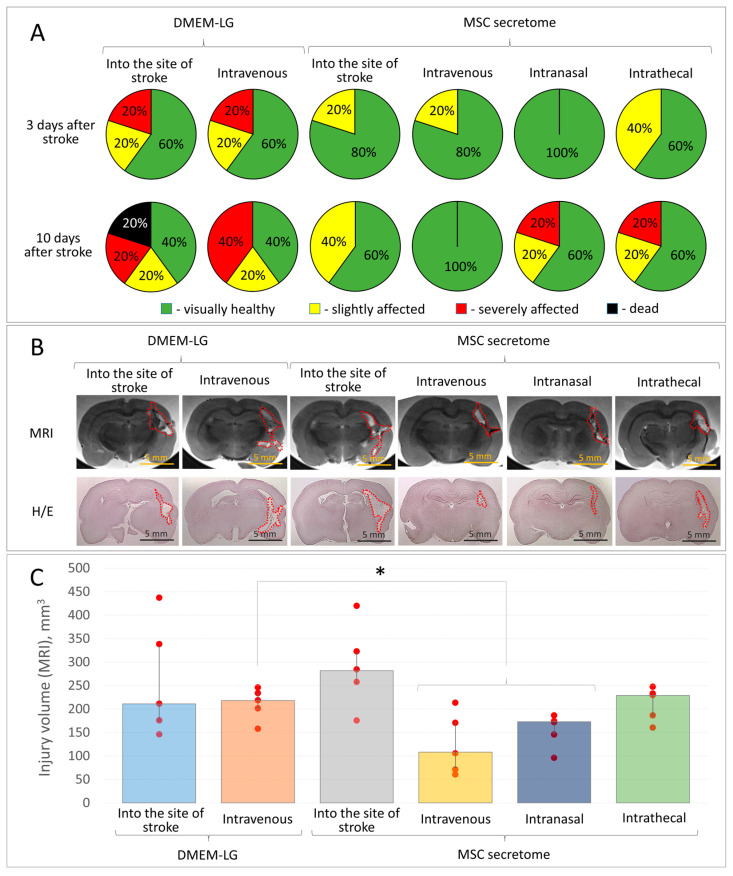
The results of in vivo studies of the neuroprotective activity of MSC secretome (10-fold concentrated) combined with various routes of administration (into the hemorrhage focus, intravenous, intranasal, intrathecal) in the model of intracerebral hemorrhage in rats: (**A**) the severity of neurological deficits in experimental animals 3 and 10 days after hemorrhage modeling; (**B**) samples of the brain lesion foci 14 days after hemorrhage modeling (MRI, histochemical staining), H/E—Hematoxylin/Eosin; (**C**) quantitative assessment of the brain lesion volume according to MRI. Data are presented as a median (25%; 75%). * *p* < 0.035, *n* = 5, Student’s *t*-test.

**Figure 2 pharmaceutics-15-01608-f002:**
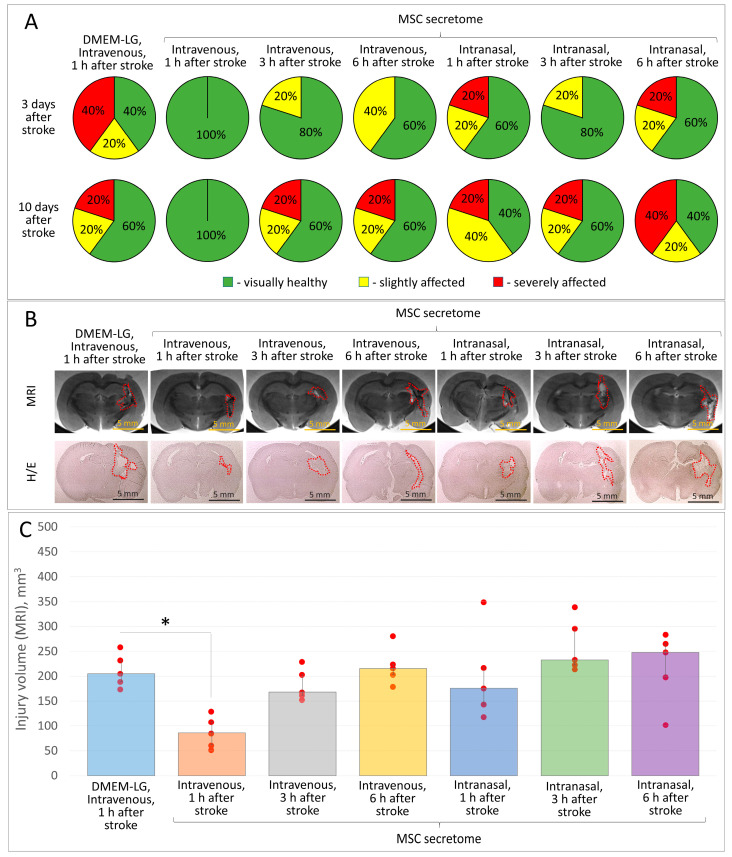
The results of in vivo studies of the neuroprotective activity of MSC secretome (10-fold concentrated) depending on the door-to-treatment time (1, 3, and 6 h after the hemorrhage modeling) using intravenous and intranasal routes of administration in the model of intracerebral hemorrhage in rats: (**A**) the severity of neurological deficits in experimental animals 3 and 10 days after hemorrhage modeling; (**B**) samples of the brain lesion foci 14 days after hemorrhage modeling (MRI, histochemical staining), H/E—Hematoxylin/Eosin; (**C**) quantitative assessment of the brain lesion volume according to MRI. Data are presented as a median (25%; 75%). * *p* ≤ 0.001, *n* = 5, Student’s *t*-test, *t* = −5.944 with 8 degrees of freedom.

**Figure 3 pharmaceutics-15-01608-f003:**
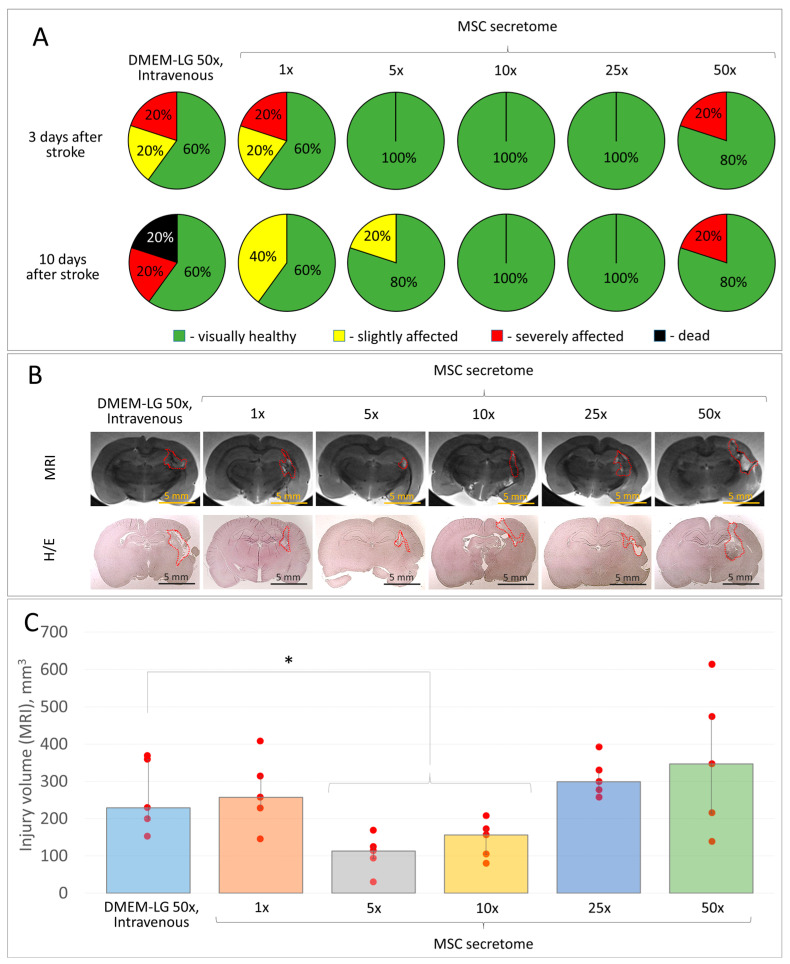
The results of in vitro studies of MSC secretome depending on the concentration used (non-concentrated 1×, 5-, 10-, 25-, and 50-fold concentrated) being intravenously administered 1 h after modeling the intracerebral hemorrhage in rats: (**A**) the severity of neurological deficits in experimental animals 3 and 10 days after hemorrhage modeling; (**B**) samples of the brain lesion foci 14 days after hemorrhage modeling (MRI, histochemical staining), H/E—Hematoxylin/Eosin; (**C**) quantitative assessment of the brain lesion volume according to MRI. Data are presented as a median (25%; 75%). * *p* < 0.05, *n* = 5, Student’s *t*-test.

**Figure 4 pharmaceutics-15-01608-f004:**
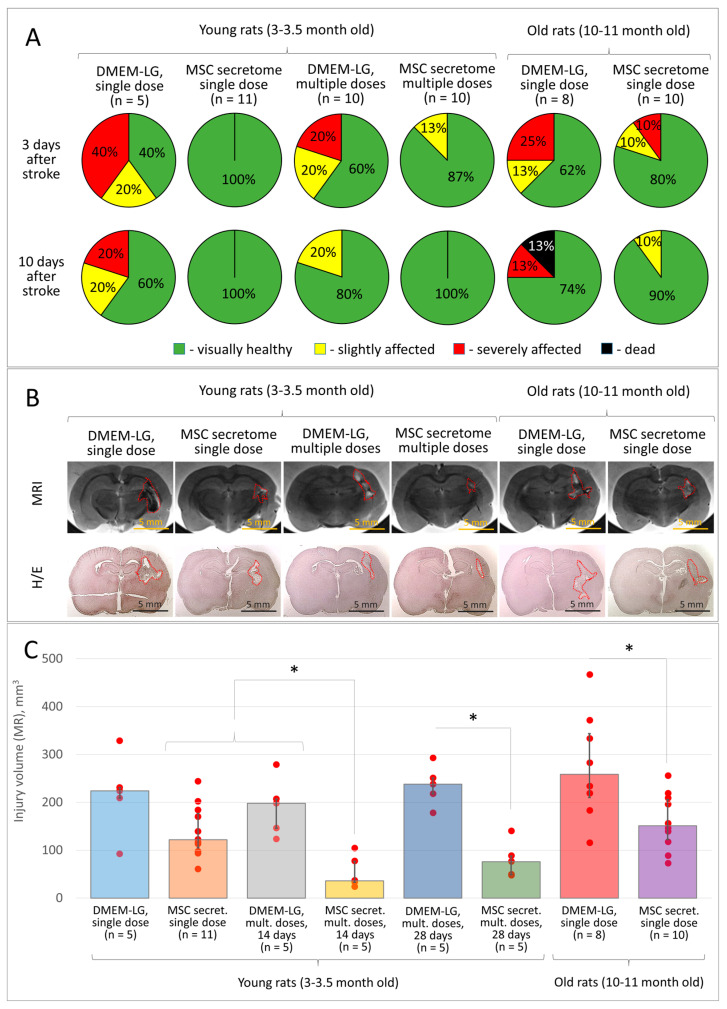
The results of in vitro studies of MSC secretome during its course administration (intravenous, 10× MSC secretome) in the model of intracerebral hemorrhage in rats: (**A**) the severity of neurological deficits in experimental animals 3 and 10 days after hemorrhage modeling; (**B**) samples of the brain lesion foci 14 days after hemorrhage modeling (MRI, histochemical staining), H/E—Hematoxylin/Eosin; (**C**) quantitative assessment of the brain lesion volume according to MRI. Data are presented as a median (25%; 75%). * *p* < 0.05, Student’s *t*-test.

## Data Availability

Data is available on request.

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
