# Peer review of "The Secretome of Mesenchymal Stromal Cells in Treating Intracerebral Hemorrhage: The First Step to Bedside"

_pharmaceutics, 2023, doi:10.3390/pharmaceutics15061608_

Round 1

Reviewer 1 Report

I thought the visual presentations of your experimental findings were very helpful.

My only problem was with the paper’s Introduction and Discussion sections where you seemed to be writing about large, usually high pressure, intracerebral bleeds that displace parts of the affected cerebral hemisphere and may compromise the brain stem’s circulation, leading to appreciable mortality, while you seem to have investigated smaller haematomas in your rats. There was no early stage mortality in the animals you studied and,  as far as I could see, no horizontal brain displacement in the coronal sections illustrated.  Possibly you might wish to take this point into consideration as the paper does not seem to mention mass effects of the blood collection.

Also. could you comment on the likely consistency of composition of future batches of your treatment preparation. This may become an issue in further studies and if the preparation, or an active component of it, comes to commercial marketing.

In the greater part of the paper the English usage is idiomatic and seems of good quality, but the first 2 sentences of the Introduction are rather chaotic and need revision. The word ‘route’ is sometimes spelled ‘root’.

Author Response

Dear Reviewer,

Thank You so much for your time and valuable comments.
Here we give the answer:

1. My only problem was with the paper’s Introduction and Discussion sections where you seemed to be writing about large, usually high pressure, intracerebral bleeds that displace parts of the affected cerebral hemisphere and may compromise the brain stem’s circulation, leading to appreciable mortality, while you seem to have investigated smaller haematomas in your rats. There was no early stage mortality in the animals you studied and,  as far as I could see, no horizontal brain displacement in the coronal sections illustrated.  Possibly you might wish to take this point into consideration as the paper does not seem to mention mass effects of the blood collection.

Yes, we agree with your fair remark that large and smaller hemorrhages may cause death of humans and animals via different mechanisms. Large hematomas are mainly lethal due to the displacement of the brain structures and compression of the brain stem. The therapeutic strategy of large hemorrhages must necessarily include surgical removal of the hematoma, and without it, therapy with a secretome (or another drug) is not effective and has little sense.
Smaller hemorrhages despite of the smaller volume of blood shed also pose a danger. They trigger neuroinflammation and secondary damage to the brain tissue, which lead to the lesion expansion, and if this affects the vital brain centers, it also may lead to the death of a person or animal. Here, in this manuscript, we consider just such an option, and as the main criteria for assessing the neuroprotective activity of the MSC secretome-based drug, we use the severity of neurological deficits and the brain lesion volume according to MRI, since they proved to be the most informative. If we compare the model we used in rats with intracerebral hemorrhage in humans, then the administration of 20 μl of blood into the rat brain (average rat brain volume ~ 2.5 ml) corresponds to a hemorrhage in a person with a volume of up to 10 ml (average human brain volume ~ 1300 ml), which is quite a lot, although, it does not lead to the displacement of the brain structures and brain stem compression. We agree that it is necessary to emphasize the aspect that we evaluate the neuroprotective activity of the MSC secretome in the model of intracerebral hemorrhage with relatively small blood volume, and we have added the corresponding fragment to the discussion. When combined with surgical removal of the hematoma, the proposed approach may also be used for extensive hematomas to reduce the severity of the consequences induced by them. As for the statistics of the prevalence of large and smaller hematomas separately, we, unfortunately, did not find such data, and all the sources we found provide data on the epidemiology of intracerebral hemorrhage (hemorrhagic stroke) as a whole, which we present in the manuscript.

2. Also, could you comment on the likely consistency of composition of future batches of your treatment preparation. This may become an issue in further studies and if the preparation, or an active component of it, comes to commercial marketing.

Yes, we agree that the composition of biological substances (secretome) produced by cells, including mesenchymal stromal cells, depends on many parameters: the health and individual characteristics of the donor, source tissue, passage of the cell line and cultivation conditions. A certain variability in the ratio of individual components between different batches of cell secretome cannot be avoided, however, most parameters can be controlled. One of the possible approaches to obtain clinically significant amounts of secretome with a relatively constant composition between batches (under relatively stable cultivation conditions) is to stabilize the producer cell line, e.g. via using immortalized (but not transformed!) MSC cultures. An important aspect of quality control of the resulting product is the control of the concentration and activity of the biological substances that make up its composition and determine its biological effect. According to our data (Lopatina et al., 2011), brain-derived neurotrophic factor (BDNF) is one of these molecules, and we plan to control its concentration in the secretome of MSCs using ELISA. All the mentioned aspects of prevention of secretome variability and quality control of its composition are currently being developed by us and will be published in future works.

3. In the greater part of the paper the English usage is idiomatic and seems of good quality, but the first 2 sentences of the Introduction are rather chaotic and need revision. The word ‘route’ is sometimes spelled ‘root’.

They were replaced by the sentence "Stroke is a severe cerebrovascular disease accompanied by the death of part of the brain that may lead to disability or death of the patient." We prefer to leave "route" instead of "root" to avoid misunderstanding.

Reviewer 2 Report

Dear Author,

The author showed the preclinical implication of mesenchymal stromal cells secretome in a rat intracerebral hemorrhage (ICH) model. The manuscript is acceptable, with minor and major comments.

Sincerely,

Minor Comments

1. I suggest editing "MSC" in the title to "mesenchymal stromal cells" for a general audience or layperson.

2. The exhibition of graphs should be improved as well as statistics.

3. Please describe the methods for injury volume measurement using MRI.

Major Comments

1. The author showed the effect of MSC secretome on the change of injury volume by MRI and H&E staining. But the author described that the MSC secretome had a neuroprotective effect. However, the decreased injury volume is limited to compensate for the direct evidence of neuroprotection. Therefore the author could provide more evidence of neuroprotection using different methods that the group previously performed (Karagyaur M et al. 21 Pharmaceutics, 2021) to make a difference between the two manuscripts.

2. Unless the author specified the effective molecules in the MSC secretome in this manuscript, the author couldn’t emphasize the novelty of this study compared to the previous publication. The author should perform the analysis of secretome, e.g., secretome ELISA or multiplex, etc.

Minor editing of English language required

Author Response

Dear Reviewer,

Thank You so much for your time and valuable comments.
Here we give the answer:

Minor Comments

1. I suggest editing "MSC" in the title to "mesenchymal stromal cells" for a general audience or layperson

Fixed.

2. The exhibition of graphs should be improved as well as statistics.

Please specify what could be improved in the graphical presentation of the results. We'd love to do it, but we don't know how. When preparing the manuscript, we tried to make the drawings as informative and easy to read as possible. Information on the statistical analysis method used was added to the figure captions.

3. Please describe the methods for injury volume measurement using MRI

The volume of brain lesion according to MRI data was estimated as the product of the sum of the areas of damaged brain tissue on individual MRI slices and the sum of the slice thickness (0.5 mm) and the distance between slices (0.75 mm). The area of the damaged brain tissue on individual MRI slices was determined using the Radiant DICOM Viewer 2020.1.1 program (Medixant, Poznan, Poland). The description of this method for determining the volume of brain lesion according to MRI data was added to the corresponding section of materials and methods.

Major Comments

1. The author showed the effect of MSC secretome on the change of injury volume by MRI and H&E staining. But the author described that the MSC secretome had a neuroprotective effect. However, the decreased injury volume is limited to compensate for the direct evidence of neuroprotection. Therefore the author could provide more evidence of neuroprotection using different methods that the group previously performed (Karagyaur M et al. 21 Pharmaceutics, 2021) to make a difference between the two manuscripts 

This work follows our previous publication (Karagyaur M et al. Pharmaceutics, 2021), where we investigated the possible mechanisms of the neuroprotective activity of MSCs. Previously, we have shown that the key mechanisms that determine the neuroprotective activity of MSC secretome are the direct neuroprotection of neural cells from glutamate-induced death (using the model of glutamate-mediated toxicity of SH-SY5Y cells in vitro) and the suppression of neuroinflammation in vivo (suppression of microglial activation) and in vitro (inhibition of monocytes/macrophages activation - not published). The aim of this work is to investigate the possibility of translating the discovered composition into clinical practice. To do this, we studied whether the neuroprotective activity of MSC secretome retains and how it depends on different routes, doses and times of administration, etc. We assume that the observed mechanism of secretome neuroprotective activity in this work is the same as in the previously described article (Karagyaur M et al. Pharmaceutics, 2021), since the composition used has a similar origin, composition, test model and method of application. The role of individual molecules (BDNF, uPA, and others) in the neuroprotective activity of the secretome in the model of intracerebral hemorrhage is currently being investigated by us and the results will be published in subsequent papers.

2. Unless the author specified the effective molecules in MSC secretome in this manuscript, the author couldn’t emphasize the novelty of this study compared to the previous publication. The author should perform the analysis of secretome, e.g., secretome ELISA or multiplex, etc. 

In our previous publication, we conducted a proof-of-concept study and showed that MSC secretome being administered intracerebrally 5 minutes after the modelling of an intracerebral hemorrhage stimulates the survival of experimental animals, reduces the severity of neurological deficits and the lesion volume according to MRI and histological studies, some of the mechanisms of its neuroprotective activity were also revealed. However, many questions remained unresolved:
1. does the secretome retain its neuroprotective activity being administered using alternative (more convenient and clinically relevant) routes of administration?
2. what dose of secretome and door-to-treatment time are optimal?
3. does the secretome reveals its neuroprotective activity in old rats?
Here we answer these questions and demonstrate that MSC secretome is a promising candidate for neuroprotection after intracerebral hemorrhages not only in laboratory but also in clinically relevant conditions - and this is its novelty. We are not aware of other studies that have explored these parameters of MSC secretome usage in intracerebral hematoma (hemorrhagic stroke). In our opinion, the range of tasks and research carried out corresponds to the profile of Pharmaceutics (MDPI).
As for the study of MSC secretome composition, these data are not new and original. Proteomic analysis of the secretome of MSCs isolated from adipose tissue has been repeatedly performed by our team and colleagues [1–3]. MSC secretome for this study was obtained from a similar cell culture (human adipose-derived mesenchymal stromal cells, 3-8 passages) using the previously described technology by the same people. During quality control of the obtained secretome, the content of BDNF (3.2 ± 1.2 ng/mL), VEGF (0.27 ± 0.13 ng/mL), HGF (0.15 ± 0.07 ng/mL), and uPA (0.34 ± 0.16 ng/mL) was determined. However, to control the composition quality of the secretome within the framework of the organization of its production, the multiparametric analysis seems to be laborious, expensive and impractical. Therefore, when producing a substance for further drug preparation, we plan to control the secretome composition quality according to the content of BDNF (e.g., using ELISA), as one of its key active substances, which determines the neuroprotective and neurotrophic effects of the secretome (our data are being prepared for publication and previously published works [4-6]). Now we develop and evaluate BDNF-based method of control of MSC secretome quality. According to our preliminary data it looks promising. Therefore, in this paper, the composition quality control of the secretome and the control of its concentration were conducted by the content of BDNF.

1. Kalinina N, Kharlampieva D, Loguinova M, Butenko I, Pobeguts O, Efimenko A, Ageeva L, Sharonov G, Ischenko D, Alekseev D, Grigorieva O, Sysoeva V, Rubina K, Lazarev V, Govorun V. Characterization of secretomes provides evidence for adipose-derived mesenchymal stromal cells subtypes. Stem Cell Res Ther. 2015 Nov 11;6:221. doi: 10.1186/s13287-015-0209-8.
2. Sagaradze GD, Grigorieva OA, Efimenko AY, Chaplenko AA, Suslina SN, Sysoeva VY, Kalinina NI, Akopyan ZhA, Tkachuk VA. [Therapeutic potential of human mesenchymal stromal cells secreted components: a problem with standartization]. Biomed Khim. 2015 Nov-Dec;61(6):750-9. Russian. doi: 10.18097/PBMC20156106750.
3. Salgado AJ, Reis RL, Sousa NJ, Gimble JM. Adipose tissue derived stem cells secretome: soluble factors and their roles in regenerative medicine. Curr Stem Cell Res Ther. 2010 Jun;5(2):103-10. doi: 10.2174/157488810791268564.
4. Lopatina T, Kalinina N, Karagyaur M, Stambolsky D, Rubina K, Revischin A, Pavlova G, Parfyonova Y, Tkachuk V. Adipose-derived stem cells stimulate regeneration of peripheral nerves: BDNF secreted by these cells promotes nerve healing and axon growth de novo. PLoS One. 2011 Mar 14;6(3):e17899. doi: 10.1371/journal.pone.0017899.
5. Ahn SY, Sung DK, Kim YE, Sung S, Chang YS, Park WS. Brain-derived neurotropic factor mediates neuroprotection of mesenchymal stem cell-derived extracellular vesicles against severe intraventricular hemorrhage in newborn rats. Stem Cells Transl Med. 2021 Mar;10(3):374-384. doi: 10.1002/sctm.20-0301.
6. Martins LF, Costa RO, Pedro JR, Aguiar P, Serra SC, Teixeira FG, Sousa N, Salgado AJ, Almeida RD. Mesenchymal stem cells secretome-induced axonal outgrowth is mediated by BDNF. Sci Rep. 2017 Jun 23;7(1):4153. doi: 10.1038/s41598-017-03592-1.

Reviewer 3 Report

The study is very interesting, well done and well written. The numbers of animals should be larger, but statistically the results seem solid.

I only highlight a few points that the authors should fix

Results

Section 3.1,3.2 the authors should indicate how concentrated is the secretome of MSCs and DMEM. Even if reported in the materials and methods they should indicate it, both in the results and in the caption related to the first two figures

Otherwise the paper seems to me to be well written and suitable for publication

Author Response

Dear Reviewer,

Thank You so much for your time, appreciation and valuable comments.
Here we give the answer:

1. Section 3.1,3.2 the authors should indicate how concentrated is the secretome of MSCs and DMEM. Even if reported in the materials and methods they should indicate it, both in the results and in the caption related to the first two figures

In these sections, we used the 10-fold concentrated MSC secretome, 10x. This information was added to Sections 3.1 and 3.2 and figure captions.

Round 2

Reviewer 2 Report

Dear Author,

Thank you for your effort to revise the manuscript with the appropriate responses. It deserves to accept the edited version.

Best,